


# Conditioning Ensemble Streamflow Prediction with the North Atlantic Oscillation improves skill at longer lead times

Seán Donegan[1], Conor Murphy[1], Shaun Harrigan[2], Ciaran Broderick[3], Saeed Golian[1], Jeff Knight[4], Tom Matthews[5], Christel Prudhomme[2,5,6], Dáire Foran Quinn[1], Adam A. Scaife[4,7], Nicky Stringer[4], and Robert L. Wilby[5]

[1]Irish Climate Analysis and Research UnitS (ICARUS), Department of Geography, Maynooth University, Co. Kildare, Ireland
[2]Forecast Department, European Centre for Medium-Range Weather Forecasts (ECMWF), Reading, UK
[3]Flood Forecasting Division, Met Éireann, Dublin 9, Ireland
[4]Met Office Hadley Centre, Exeter, UK
[5]Department of Geography and Environment, Loughborough University, Loughborough, UK
[6]UK Centre for Ecology & Hydrology (UKCEH), Wallingford, UK
[7]College of Engineering, Mathematics, and Physical Sciences, University of Exeter, Exeter, UK

**Correspondence:** Seán Donegan (sean.donegan@mu.ie)

**Abstract.** Skilful hydrological forecasts can benefit decision-making in water resources management and other water-related sectors that require long-term planning. In Ireland, no such service exists to deliver forecasts at the catchment scale. In order to understand the potential for hydrological forecasting in Ireland, we benchmark the skill of Ensemble Streamflow Prediction (ESP) for a diverse sample of 46 catchments using the GR4J hydrological model. Skill is evaluated within a 52-year hindcast

study design over lead times of 1 day to 12 months for each of 12 initialisation months, January to December. Our results show that ESP is skilful against a probabilistic climatology benchmark in the majority of catchments up to several months ahead. However, the level of skill was strongly dependent on lead time, initialisation month, and individual catchment location and storage properties. Mean ESP skill was found to decay rapidly as a function of lead time, with continuous ranked probability skill scores of 0.8 (1-day), 0.32 (2-week), 0.18 (1-month), 0.05 (3-month), and 0.01 (12-month). Forecasts were generally more

skilful when initialised in summer than other seasons. A strong correlation ($\rho = 0.94$) was observed between forecast skill and catchment storage capacity (baseflow index), with the most skilful regions, the Midlands and East, being those where slowly responding, high storage catchments are located. Results also highlight the potential utility of ESP for decision-making, as measured by its ability to forecast low and high flow events. In addition to our benchmarking experiment, we conditioned ESP on the winter North Atlantic Oscillation (NAO) using adjusted hindcasts from the Met Office's Global Seasonal Forecasting

System version 5. We found gains in winter forecast skill of 7–18% were possible over lead times of 1 to 3 months, and that NAO-conditioned ESP is particularly effective at forecasting dry winters, a critical season for water resources management. We conclude that ESP is skilful in a number of different contexts and thus should be operationalised in Ireland given its potential benefits for water managers and other stakeholders.



# 1 Introduction

Skilful hydrological forecasts at lead times of weeks to months can benefit water resources management (Anghileri et al., 2016; Dixon and Wilby, 2019; Viel et al., 2016; Wetterhall and Di Giuseppe, 2018) and help mitigate extreme events by enhancing preparedness and improving operational decisions (Luo and Wood, 2007; Neumann et al., 2018; Pappenberger et al., 2015a; Zhao and Zhao, 2014). For example, hydrological forecasts have been used to modify reservoir operations for hydropower production (Fan et al., 2016), storage and supply (Turner et al., 2017), and the management of flood and drought

conditions (Amnatsan et al., 2018; Ficchì et al., 2016; Watts et al., 2012). They have also been shown to benefit sectors such as agriculture (Mushtaq et al., 2012), tourism (Fundel et al., 2013), and navigation (Meißner et al., 2017). Such applications can yield significant economic returns. For instance, Hamlet et al. (2002) reported a potential rise in annual revenue of $153 million when forecast information was incorporated into the operation of major hydropower dams in the Columbia River basin. Similarly, Pappenberger et al. (2015a) claim that the European Flood Awareness System (EFAS; Thielen et al., 2009) saves

around 400 Euro for every 1 Euro invested.

The value of hydrological forecasting has led several countries to establish operational seasonal hydrological forecasting (SHF) systems. These include the U.S. National Weather Service's (NWS) Hydrologic Ensemble Forecast Service (HEFS; Demargne et al., 2014), the Hydrological Outlook UK (HOUK; Prudhomme et al., 2017), and the Australian Bureau of Meteorology's statistical and dynamical forecasts (Schepen and Wang, 2015). Although Ireland benefits from regional hydrological outlooks

provided by EFAS, no service currently exists for delivering forecasts at the catchment scale; yet water managers and other stakeholders require confident, locally-tailored forecast information. A national operational SHF system could bridge this gap. However, despite interest from water managers, it is difficult to justify the implementation of such a system as little preparatory work has been done to evaluate the potential for hydrological forecasting in an Irish context.

Recent international assessments of progress in SHF (Tang et al., 2016; Yuan et al., 2015) indicate that: (i) advances in empirical

and dynamical SHF are feasible in climate contexts that resemble Ireland; and (ii) SHF spans a wide range of methods with varying complexity and data requirements, but no universally accepted 'best' approach has emerged. As the performance of different methods will likely depend on time of year, lead time, and, critically, local hydrological context (Girons Lopez et al., 2020; Harrigan et al., 2018; Meißner et al., 2017; Pechlivanidis et al., 2020), understanding how best to apply the range of available tools to develop skilful forecasts for Ireland requires rigorous testing at the catchment scale. To the authors'

knowledge, only Foran Quinn et al. (accepted) have previously evaluated seasonal streamflow forecasts for Ireland. They found that whilst skill was mainly restricted to summer months, statistical persistence forecasts could have practical value in the management of water resources and hydrological extremes. We build on this work by assessing the scientific basis for SHF in Ireland by evaluating and benchmarking the skill of Ensemble Streamflow Prediction (ESP).

ESP is a well-established forecasting technique in which historical sequences of climate data at the time of forecast are used

to drive a hydrological model, initialised with current land surface conditions, producing an ensemble of equiprobable future streamflow traces (Day, 1985; Twedt et al., 1977). ESP is comparable to persistence in that it requires no information about





future meteorological conditions; outlooks are instead based on knowledge of hydrological state variables (i.e., antecedent soil moisture, groundwater, snowpack, and streamflow itself) which can provide predictability up to 5 months ahead (Wood and Lettenmaier, 2008). In this regard, ESP can be used to efficiently specify not only the catchments where knowledge of initial

conditions or meteorological forcing may be the greatest source of skill, but also the time of year and lead times over which different skill sources may be dominant (Wood and Lettenmaier, 2006).

The ESP method was originally developed in the snow-dominated catchments of the western United States (e.g., Franz et al., 2003), but has shown skill in other regions, including the UK (Harrigan et al., 2018), European Alps (Förster et al., 2018), Sweden (Girons Lopez et al., 2020), New Zealand (Singh, 2016), Australia (Pagano et al., 2010; Wang et al., 2011), and China

(Yuan et al., 2016). Simplicity and efficiency make ESP a popular choice for operational forecasting. It is one of three methods used in the HOUK (Prudhomme et al., 2017) and forms the basis of the NWS HEFS (Demargne et al., 2014). Moreover, ESP is recognised as a low-cost, 'tough-to-beat' forecast (Pappenberger et al., 2015b) against which value-added by more sophisticated hydrometeorological ensemble systems can be assessed (e.g., Arnal et al., 2018; Bazile et al., 2017; Wanders et al., 2019). Hence, the potential application of ESP in Ireland merits exploration.

However, lack of sensitivity to concurrent meteorological conditions limits the application of ESP in areas that are less dependent on initial hydrological conditions. Given that local meteorological conditions are known to be teleconnected to regional variations in atmospheric–oceanic modes, ESP techniques may be improved by conditioning on these circulation patterns. This reduces ensemble spread by eliminating those scenarios which are less likely to occur given the large-scale synoptic situation. Several studies have already demonstrated the added value of incorporating climate information into ESP forecasts in this way.

For example, Hamlet and Lettenmaier (1999) found that conditioning ESP traces according to El Niño–Southern Oscillation (ENSO) and Pacific Decadal Oscillation indicators significantly improved forecast specificity and extended lead time by about six months in the Columbia River basin. Similarly, both Werner et al. (2004) and Bradley et al. (2015) reported improvements of 28% and 27% in forecast skill, respectively, when conditioning ESP with ENSO. More modest improvements of 5–10% were observed by Beckers et al. (2016) for two test stations when applying an ENSO-conditioned ESP.

In Europe, the dominant mode of climate variability is the North Atlantic Oscillation (NAO). The NAO affects streamflow predictability, particularly during winter (Bierkens and van Beek, 2009; Steirou et al., 2017; Wedgbrow et al., 2002; Wilby, 2001), and is highly correlated with winter streamflow over Ireland (Murphy et al., 2013). As winter is the most important season for groundwater recharge in Europe, the ability to accurately forecast winter streamflow would be extremely beneficial for water managers. Advances in predicting the NAO (Scaife et al., 2014; Smith et al., 2020) enable long-range forecasts of UK

winter hydrology (Svensson et al., 2015) as well as improved seasonal meteorological forecasts for driving hydrological models (Stringer et al., 2020). Hence, it may be possible to leverage this predictability to improve ESP performance by sub-sampling ensemble members for Ireland using the winter NAO.

In this paper, we follow the approach of Harrigan et al. (2018) by benchmarking ESP skill against streamflow climatology within a 52-year hindcast study design, using various lead times and initialisation months, and for diverse catchment types.





We also examine the effect of conditionally sampling ensemble members on ESP skill during winter months. The following research questions are addressed:

1. When is ESP skilful, given a wide range of lead times and initialisation months?

2. Where is ESP most skilful, at regional and catchment scales?

3. How does ESP skill relate to catchment characteristics?

4. To what extent can winter ESP skill be improved by conditioning on the NAO?

5. What is the potential for operationalising the ESP method for hydrological forecasting in Ireland?

Section 2 describes our data and methods. Our results are presented in Sect. 3. We offer discussion and suggestions for future research in Sect. 4. Conclusions are presented in Sect. 5.

## 2 Data and methods

### 2.1 Catchment selection and observed data

Forty-six catchments were selected for our analysis following the same criteria used to establish the Irish Reference Network (Murphy et al., 2013). Catchments were selected provided they: (i) had quality-assured, long-term observational data, with a minimum record length of 25 years; (ii) had a flow regime which had not been significantly altered by human activity; (iii) had little evidence of land-use change; and (iv) together build a representative sample of Ireland's diverse hydrological

and climatological conditions, with good spatial coverage. This selection process ensured sufficient data for hydrological model calibration whilst limiting the potential for confounding factors that could adversely affect the interpretation of results. Catchments were grouped according to the European Union's NUTS (Nomenclature of Territorial Units for Statistics) III regions (Fig. 1) for spatial analysis. As the Dublin region contained only one catchment in our sample, this was merged with the Mid-East into a single region: The East. This yielded regions with between four to ten study catchments in each.

Observed daily mean streamflow data ($m^3\,s^{-1}$) were obtained from gauging stations administered by the Office of Public Works (OPW) and the Environmental Protection Agency. Despite the strict selection criteria, some catchments still contain multiple or extended periods of missing data. Hence, streamflow records were retrieved only for calendar years 1992–2017 – the longest 'usable' period common to all 46 catchments. Catchment average daily precipitation ($mm\,d^{-1}$) and temperature (°C) spanning 1961–2017 were derived from gridded (1 km × 1 km) datasets developed by Met Éireann (Walsh, 2012). Potential evaporation

($mm\,d^{-1}$) was calculated from temperature and radiation according to Oudin et al. (2005).

Data on catchment physical attributes were based on a selection of physical catchment descriptors (PCDs) from the OPW's Flood Studies Update (Mills et al., 2014). These PCDs describe facets of catchment hydrology, morphology, soil, and climate, and are used here to examine relationships between catchment characteristics and ESP skill. The primary PCDs of interest are the baseflow index (BFI), the Richards–Baker flashiness index (RBI; Baker et al., 2004), and the runoff ratio (RR), as





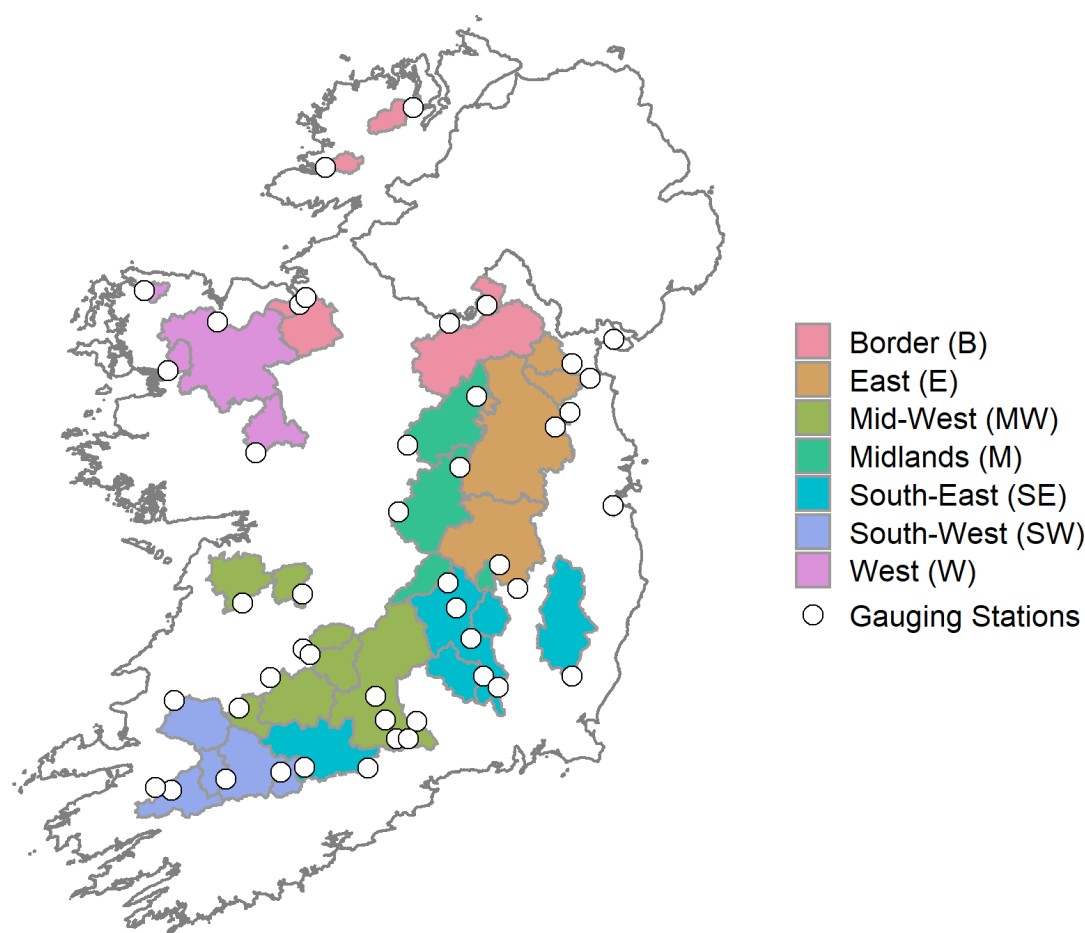

**Figure 1.** Location of the 46 study catchments, shaded by region, and associated gauging stations (white dots).

these describe aspects of catchment storage and response, which can significantly affect ESP skill (e.g., Girons Lopez et al., 2020; Harrigan et al., 2018; Pechlivanidis et al., 2020). The BFI is calculated according to the Institute of Hydrology method (Gustard et al., 1992) and quantifies the contribution of stored sources to runoff. Hence, the BFI can be considered an integrated measure of catchment storage capacity. The RBI measures the frequency and rapidity of short-term changes in streamflow, and the RR gives the amount of runoff relative to the amount of precipitation received. Across the sample of catchments, median (5[th] and 95[th] percentile) BFI is 0.59 (0.34, 0.75), median RBI is 0.19 (0.07, 0.5), and median RR is 0.62 (0.5, 0.82). Higher values of RBI and RR are observed for catchments with lower storage capacity (BFI) and smaller area, indicative of more responsive hydrological regimes. In addition to the BFI, we also represent catchment storage using the calibrated GR4J $x_1$ and $x_3$ parameters, the sum of which give an overall indicator of storage capacity. A complete list of PCDs referred to in this study is given in Table 1. Catchment characteristics are summarised for Ireland and each of the NUTS III regions in Table 2, and for individual catchments in Table S1 in the Supplement.





**Table 1.** Physical catchment descriptors referred to in this study.

| Descriptor | Explanation | Units | Range |
|---|---|---|---|
| BFI | Baseflow index; proportion of runoff derived from stored sources | – | 0–1 |
| RBI | Richards–Baker flashiness index; oscillations in flow relative to total flow | – | 0–1 |
| RR | Runoff ratio; ratio of runoff to received precipitation | – | 0–1 |
| AREA | Catchment area | $km^2$ | – |
| SAAR | Standard-period (1961–1990) average annual rainfall | mm | – |
| FLATWET | Proportion of time soils expected to be typically quite wet | – | 0–1 |
| PEAT | Proportional extent of catchment area classified as peat bog | – | 0–1 |
| FOREST | Proportional extent of forest cover | – | 0–1 |
| MSL | Main-stream length | km | – |
| S1085 | Slope of main stream excluding the bottom 10% and top 15% of its length | $m\,km^{-1}$ | – |
| TAYSLO | Taylor–Schwartz measure of mainstream slope | $m\,km^{-1}$ | – |

**Table 2.** Summary statistics of seven catchment characteristics for Ireland and each NUTS III region. The median across $n$ catchments is given with the 5[th] and 95[th] percentile ranges in parentheses. Mean annual runoff ($\overline{Q}$), precipitation ($\overline{P}$), and potential evaporation ($\overline{PE}$) were calculated over calendar years 1992–2017.

| Region | $n$ | Area ($km^2$) | $\overline{Q}$ ($mm\,yr^{-1}$) | $\overline{P}$ ($mm\,yr^{-1}$) | $\overline{PE}$ ($mm\,yr^{-1}$) | BFI (–) | RBI (–) | RR(–) |
|---|---|---|---|---|---|---|---|---|
| IE | 46 | 412 | 686 | 1149 | 565 | 0.59 | 0.19 | 0.62 |
| | | (23, 2286) | (431, 1336) | (905, 1861) | (529, 580) | (0.34, 0.75) | (0.07, 0.5) | (0.5, 0.82) |
| B | 6 | 180 | 970 | 1484 | 540 | 0.43 | 0.24 | 0.73 |
| | | (94, 1279) | (569, 1371) | (1088, 1878) | (521, 551) | (0.3, 0.72) | (0.07, 0.55) | (0.59, 0.83) |
| E | 8 | 290 | 483 | 926 | 560 | 0.62 | 0.15 | 0.55 |
| | | (7, 2193) | (385, 750) | (891, 1149) | (535, 574) | (0.44, 0.72) | (0.12, 0.45) | (0.47, 0.7) |
| MW | 10 | 606 | 697 | 1177 | 571 | 0.58 | 0.2 | 0.64 |
| | | (225, 1891) | (506, 900) | (1043, 1373) | (561, 585) | (0.45, 0.67) | (0.09, 0.36) | (0.5, 0.75) |
| M | 6 | 360 | 524 | 986 | 561 | 0.71 | 0.13 | 0.56 |
| | | (38, 1147) | (440, 644) | (914, 1125) | (556, 566) | (0.53, 0.8) | (0.08, 0.26) | (0.52, 0.62) |
| SE | 6 | 738 | 644 | 1085 | 567 | 0.56 | 0.26 | 0.58 |
| | | (145, 2397) | (473, 1044) | (981, 1325) | (545, 576) | (0.42, 0.66) | (0.19, 0.45) | (0.51, 0.85) |
| SW | 6 | 603 | 929 | 1581 | 569 | 0.44 | 0.4 | 0.71 |
| | | (269, 1206) | (668, 1500) | (1417, 1987) | (567, 574) | (0.34, 0.61) | (0.13, 0.5) | (0.65, 0.8) |
| W | 4 | 308 | 1046 | 1512 | 552 | 0.6 | 0.18 | 0.7 |
| | | (87, 1749) | (723, 1223) | (1198, 1695) | (545, 563) | (0.32, 0.75) | (0.1, 0.54) | (0.63, 0.76) |





## 2.2 Hydrological modelling

The GR4J (Génie Rural à 4 paramètres Journalier; Perrin et al., 2003) daily lumped conceptual rainfall-runoff model was applied. This model has a parsimonious structure consisting of four free parameters ($x_1$–$x_4$) that require calibration of observed streamflow data against precipitation and potential evaporation. The model structure can be described in terms of its water

balance and routing operators (Santos et al., 2018). Water is partitioned between a production (soil moisture accounting) store and a routing store. The production store (capacity $x_1$ mm) gains water from rainfall and loses water from evaporation and percolation. Ninety percent of the total quantity of water reaching the routing component (i.e., the sum of the percolation leak and the water bypassing the production store) is routed by a single unit hydrograph (time base $x_4$ d) and a non-linear routing store (capacity $x_3$ mm). The remaining 10% is routed by a single unit hydrograph (time base $2(x_4)$ d). A groundwater exchange

function (rate $x_2$ mm d$^{-1}$) operates on both routing channels and can be positive, negative, or zero.

We chose GR4J on the basis of its reliability. The model has undergone extensive testing in several countries and has been shown to accurately simulate the hydrology of diverse catchment types, with comparatively good results (e.g., Coron et al., 2012; Perrin et al., 2003; Vaze et al., 2011). It has also been successfully applied to Irish conditions (Broderick et al., 2016, 2019) where it was found to perform well for a similar set of catchments to those used here, with respect to both temporal

transition between contrasting climate periods and the reproduction of various hydrological signatures. Moreover, GR4J has been used previously for ESP (Harrigan et al., 2018; Pagano et al., 2010). We find the model uniquely suited to this application, as large ensembles of runs are required in long hindcast experiments. These simulations can be computationally intensive and time consuming with more complex model structures, which do not necessarily lead to large improvements in skill (e.g., Bell et al., 2017). GR4J is implemented in R via the open-source 'airGR' package (v.1.4.3.65; Coron et al., 2017, 2020).

Model parameters were estimated using Memetic Algorithms with Local Search Chains (MA-LS-Chains; Bergmeir et al., 2016; Molina et al., 2010). As ESP forecasts are made throughout the year under varying conditions, the non-parametric Kling–Gupta efficiency (KGE$_{NP}$; Appendix A) was chosen as the objective function to optimise, as it has been shown to capture multiple parts of the hydrograph well (Pool et al., 2018). Parameter estimation was carried out in R using the 'Rmalschains' package (v.0.2-6 Bergmeir et al., 2016, 2019) with the Covariance Matrix Adaptation Evolution Strategy (Hansen and Ostermeier, 2001)

as the local search method.

Model calibration was performed following the procedures recommended by Arsenault et al. (2018). A split-sample test (Klemeš, 1986) was first used to assess model robustness. The available record was divided into two periods of equal length, denoted here as period 1 (P1; 1 January 1993–2 July 2005) and period 2 (P2; 2 July 2005–31 December 2017). Separate parameter sets were created using data from P1 and P2 in turn for calibration and validation (i.e., parameters were calibrated on P1 and val-

idated on P2 and vice versa). A third round of calibration was then performed using data from the complete period (CP; 1 January 1993–31 December 2017). This parameter set was carried forward for all subsequent modelling tasks. An approach of this nature is beneficial as it allows for evaluation of the model's ability to accurately simulate catchment processes over two independent periods whilst maximising the information content of the parameter set that is used to generate the ESP hindcast



time series. In all cases, 1992 was used as a warm-up period to initialise model states, and the full series (1993–2017) was

simulated before calibration and testing to preserve the internal dynamics and temporal stability of catchment stores. Model

performance was evaluated using $KGE_{NP}$, the Nash–Sutcliffe efficiency (NSE; Nash and Sutcliffe, 1970), and the percent bias

(PBIAS; Gupta et al., 1999).

## 2.3    ESP study design

### 2.3.1    Historical ESP

Forecasts were initialised on the first day of each month following a 4-year model warm-up period to estimate initial hydrolog-

ical conditions. The first usable forecast date after model warm-up is, therefore, 1 January 1965. For each forecast initialisation

date, a 55-member ensemble $m$ of streamflow hindcasts was generated by forcing GR4J with corresponding historic climate

sequences (pairs of precipitation and potential evaporation) extracted from 1961–2016 out to a 12-month lead time. Following

Harrigan et al. (2018), streamflow at a given lead time is expressed as the mean daily streamflow from the forecast initialisa-

tion date to $n$ days or months ahead in time. For example, a January forecast with a lead time of 1 month is the mean daily

streamflow from 1 January to 31 January, and a January forecast with a lead time of 2 months is the mean daily streamflow

from 1 January to 28 February. Average flow values are used, particularly at monthly time scales, because these are preferred

by decision-makers in many water sectors (Arnal et al., 2018). Hindcast time series were therefore temporally aggregated to

provide predictions of mean streamflow over lead times of 1 day to 12 months, resulting in 365 lead times per forecast (ex-

cluding leap days). In order to mimic operational conditions and prevent artificial skill inflation (see Robertson et al., 2016),

we also employed leave-one-out cross-validation (L1OCV) whereby data from the forecast year was not used as input to the

model, as this would not be available in a real-time forecasting setting. For example, a forecast initialised on 1 January 1965

will use historic climate sequences of 365 days in length (1 January to 31 December) extracted from 1961–2016, but not 1965.

ESP skill is evaluated over 52 initialisation years $N$ (1965–2016) with 12 initialisation months $i$ (January to December). In

total, 624 hindcasts were generated ($N \times i$) with 34,320 individual ensemble members ($N \times i \times m$), each at 365 lead times

across 46 catchments, resulting in a hindcast archive of more than $5.7 \times 10^8$ streamflow values.

### 2.3.2    Conditioned ESP

To investigate the potential for improving winter streamflow predictability, we conditioned the ESP method using adjusted

NAO hindcasts from the Met Office's Global Seasonal Forecasting System version 5 (GloSea5; MacLachlan et al., 2015).

GloSea5 is built around the high-resolution Hadley Centre Global Environmental Model version 3 (HadGEM3) which inte-

grates atmosphere, ocean, land, and sea-ice components. HadGEM3 has an atmospheric resolution of 0.83° longitude by 0.55°

latitude with 85 vertical levels and an ocean resolution of 0.25° in both latitude and longitude with 75 vertical levels. Although

GloSea5 has been shown to skilfully predict the NAO (Scaife et al., 2014), several studies have documented a signal-to-noise

problem that limits the usefulness of forecasts to drive hydrological models, as ensemble mean signals in NAO forecasts are

anomalously weak (Eade et al., 2014; Scaife et al., 2014; Scaife and Smith, 2018). Focusing on the dynamical signals can





correct this by amplifying the ensemble mean (Baker et al., 2018), so adjusted hindcasts are used here following the method of Stringer et al. (2020). For each DJF period over 1993–2015, we combined GloSea5 hindcasts initialised on 1, 9, and 17 November to create a 51-member ($3 \times 17$ members) ensemble of raw NAO predictions. After adjustment to remove the signal-to-noise discrepancy in the raw ensemble, predicted monthly NAO values were used to select 10 non-sequential DJF analogues

(e.g., December 2007, January 1980, February 2011) where the mean observed seasonal NAO approximated the mean adjusted seasonal NAO hindcast. This resulted in a 510-member ensemble of analogue date sequences which were then used to extract corresponding precipitation and potential evaporation for input to the ESP method. The decision to construct analogue seasons with months from different years was made to: (a) ensure that the range of possible values suggested by GloSea5 could be reproduced; and (b) to avoid underestimating extreme seasonal NAO values, which would sample exclusively from DJF 2009–

10 if below $-10\,\mathrm{hPa}$ (Stringer et al., 2020). Ten analogues were sampled per hindcast member to minimise non-NAO-related variability whilst keeping a consistent NAO signal across the sample. Conditioned ESP forecasts were only initialised on 1 December.

## 2.4 Skill evaluation

### 2.4.1 Hindcast performance

We quantify the overall skill of the ESP method using the continuous ranked probability score (CRPS; Hersbach, 2000) and corresponding skill score (CRPSS; Appendix B). The CRPS is a recommended and widely-used evaluation metric for ensemble hydrological forecasting (Pappenberger et al., 2015b) which penalises forecasts that are biased or have low sharpness (Wilks, 2011). For skill evaluation, we used model simulations in place of observations, generated by forcing GR4J over 1965–2016 with a 4-year warm-up period. This is common practice (e.g., Arnal et al., 2018; Harrigan et al., 2018; Wood and Lettenmaier,

2008; Wood et al., 2016) as it isolates loss of skill due to errors in initial conditions rather than model or data errors. For historical ESP, the reference forecast is constructed as the full-sample climatological distribution of modelled observations over 1965–2016 for the forecast period. L1OCV was also applied to this ensemble. Conditioned ESP is compared against both the probabilistic climatology benchmark and the full historical ESP ensemble. In all cases, the Ferro et al. (2008) ensemble size correction for CRPS is applied after cross-validation to account for differences in the number of ensemble members.

### 2.4.2 Hindcast potential utility

We further evaluate the performance of ESP using the receiver operator characteristic (ROC) score as a measure of hindcast potential utility. The ROC score describes the ability of a forecasting system to distinguish between events and non-events. It is defined as the area under the ROC curve, which plots the probability of detection against the probability of false detection for a given event and a range of probability levels (Demargne et al., 2014). The ROC score shows how well a forecasting system

predicts the correct category of an event. A ROC score of 1 indicates that all ensemble members correctly predicted the event in all years, whereas a ROC score of 0.5 indicates a forecast with no discrimination. For each initialisation month and lead time, we calculated the ROC score using the lower and upper terciles of the corresponding model simulations as thresholds





(i.e., below- and above-average streamflow categories). Hence, the ROC score should be interpreted as a measure of how well ESP can forecast low and high flow events. We use a slightly higher skill threshold of 0.6 so that forecasts are only considered

skilful if they are better than guesswork. Both CRPSS and ROC scores were calculated in R using the 'easyVerification' package (v0.4.4; MeteoSwiss, 2017).

## 3   Results

### 3.1   Hydrological model performance

GR4J performed well for our catchment sample (Fig. 2). The median ($5^{th}$ and $95^{th}$ percentile) value of $KGE_{NP}$ is 0.95 (0.88,

0.97) for calibration over P1, P2, and CP. Median validation scores of 0.91 (0.84, 0.96) were achieved during testing on both P1 and P2. Median NSE for calibration over CP is 0.88 (0.69, 0.93) and median PBIAS is 0.03% (−0.04%, 0.14%). Performance metrics and calibrated parameter values for individual catchments over CP are given in Table S1.

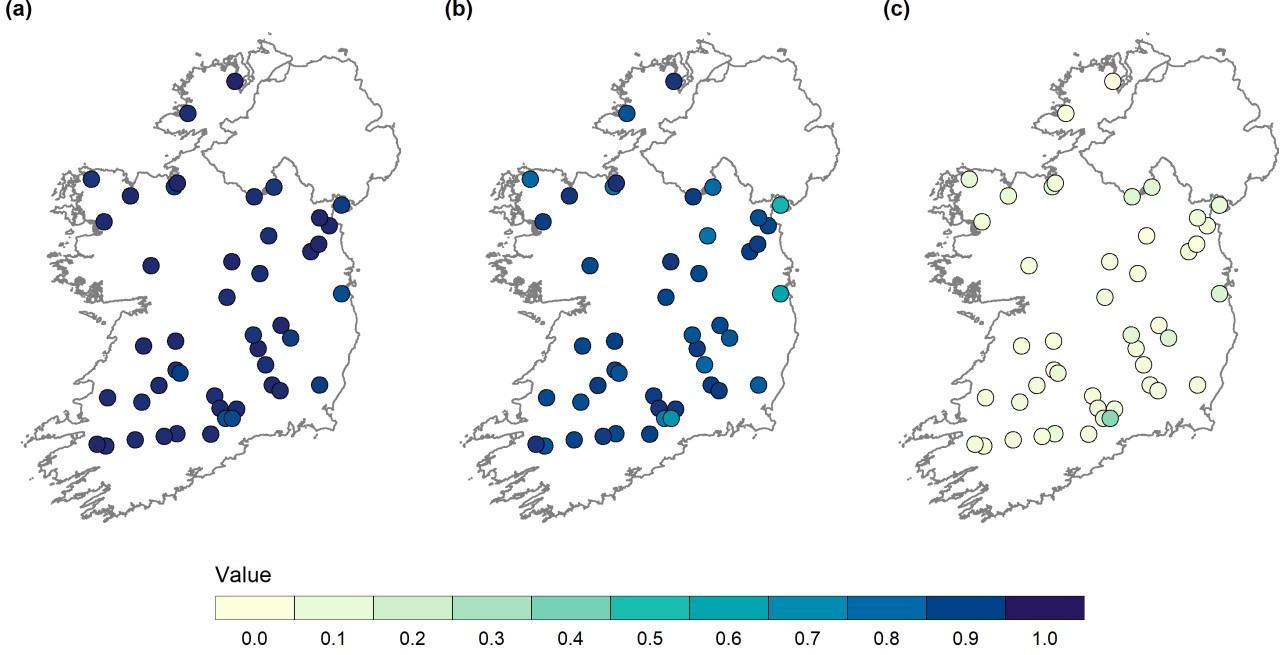

**Figure 2.** GR4J model performance over the complete period (1993–2017) as measured by (a) $KGE_{NP}$, (b) NSE, and (c) absolute PBIAS.

### 3.2   Timing of ESP skill

#### 3.2.1   Lead time

Mean ESP skill declines rapidly as a function of lead time, across all catchments and initialisation months (Fig. 3). Mean CRPSS values for short (1-day) to extended (2-week) lead times range from 0.8 to 0.32, and for monthly (1- and 2-month),





seasonal (3-month), and annual lead times from 0.18, 0.09, and 0.05, to 0.01, respectively. However, the rate at which skill

decays across catchments varies, with considerable differences around the mean shown by the 5$^{th}$ and 95$^{th}$ percentile bands.

For example, for a 2-week lead time CRPSS values within this band range between 0.1–0.58, and for a 1-month lead time

between 0.03–0.4.

**Figure 3.** Mean ESP CRPSS values across all 46 study catchments, 12 forecast initialisation months, and all 365 lead times, with short and extended lead times shown inset for readability. Variations in skill scores across all catchments at each lead time are given by the 5$^{th}$ and 95$^{th}$ percentile ensemble range.

### 3.2.2    Initialisation month

ESP skill varies with forecast initialisation month and time of year, with highest and lowest skill scores dependent on lead

time (Fig. 4). For short to monthly lead times, skill scores are highest when forecasts are initialised in summer (JJA), with

July the most skilful initialisation month on average, whereas skill tends to be lower during winter (DJF), with January and

December exhibiting the lowest skill. At seasonal lead times, skill during autumn (SON) is comparable to that of summer,





whilst the least skilful forecasts are produced in the spring months (MAM). As in Fig. 3, skill tends toward zero as lead time
increases, regardless of initialisation month. Although this decline in performance is less severe for summer than for other
seasons, by a 12-month lead time nearly all forecasts are less skilful than climatology. Despite this, several catchments have
above (below) average skill scores with some performing notably better (worse) across different lead times and initialisation
months. For example, ESP forecasts initialised in July with a 1-month lead have moderate skill on average (CRPSS = 0.34),
but seven catchments have high skill (CRPSS ≥ 0.5) with a maximum CRPSS of 0.68 for the Erkina (ID 15005). Conversely,
14 catchments have low skill (CRPSS ≤ 0.25) with a minimum of −0.03 for the Newport (ID 32012).

**Figure 4.** As in Fig. 3, but for each forecast initialisation month. Data from Fig. 3 is included in the background of each panel for reference.





### 3.3 Spatial distribution of ESP skill

#### 3.3.1 NUTS III regions

Mean ESP skill across all initialisation months is shown in Fig. 5 for Ireland and each of the seven NUTS III regions. The Midlands, Mid-West, and East are the most skilful regions, followed by the South-East, West, and Border regions. The South-West is the least skilful region on average, with the lowest CRPSS values for all sampled lead times. Regional variations in skill are less pronounced at shorter lead times but become more apparent as lead time increases. For example, at a 1-month lead time the Midlands (CRPSS = 0.26) is twice as skilful as the Border (CRPSS = 0.13) and South-West (CRPSS = 0.12).

All regions are, on average, skilful out to a 1-month lead time, but the Midlands is the only region that is moderately skilful (CRPSS ≥ 0.25). The Midlands remains the most skilful region beyond 1-month, though the level of skill is generally quite low for all regions by this point. The regional variations observed in Fig. 5 are partly explained by the relationship between catchment characteristics and ESP skill (Sect. 3.4) as the pattern is broadly consistent with differences in catchment storage capacity and wetness. For instance, the Midlands has a high median BFI of 0.71 and a low median SAAR of 939 mm, whereas

the South-West has a low median BFI of 0.44 and a high median SAAR of 1407 mm.

#### 3.3.2 Catchment scale

Notable sub-regional heterogeneity emerges when examining skill scores for individual forecasts at the catchment scale (Fig. 6). This heterogeneity is more noticeable at monthly to seasonal lead times, where skilful forecasts are possible for several catchments at different times of the year, even if average skill for the region as a whole tends to be low. For example, whilst

the South-West is the least skilful region at a 1-month lead time, with an average CRPSS of 0.12, forecasts with above average skill are possible in several catchments in the region in June, such as the Blackwater (ID 18003; CRPSS = 0.25) and the Laune (ID 22035; CRPSS = 0.22).

### 3.4 Relationship with catchment characteristics

Figure 7 shows the relationship between ESP skill, as represented by the average 1-month CRPSS, and several PCDs for each

of the 46 study catchments using the non-parametric Spearman rank correlation coefficient ($\rho$). ESP skill is closely linked with catchment storage properties and responsiveness. There are strong positive correlations between modelled storage capacity ($x_1$ + $x_3$) and BFI ($\rho = 0.8$) and between ESP skill and BFI ($\rho = 0.94$). There is also a strong positive correlation between ESP skill and modelled storage capacity ($\rho = 0.75$). Conversely, there is a strong negative correlation between ESP skill and the RBI ($\rho = -0.82$) and a moderate negative correlation between ESP skill and the RR ($\rho = -0.63$). In general, ESP skill tends to

be higher for slower responding catchments with greater storage capacity, and lower for faster responding, flashy catchments with poor infiltration. ESP skill is also positively correlated with catchment area ($\rho = 0.5$) and main-stream length ($\rho = 0.46$), indicating a tendency for the method to perform better in larger catchments with longer streams. Negative correlations exist between ESP skill and PCDs related to catchment wetness (SAAR, FLATWET, and PEAT), though these PCDs also exhibit




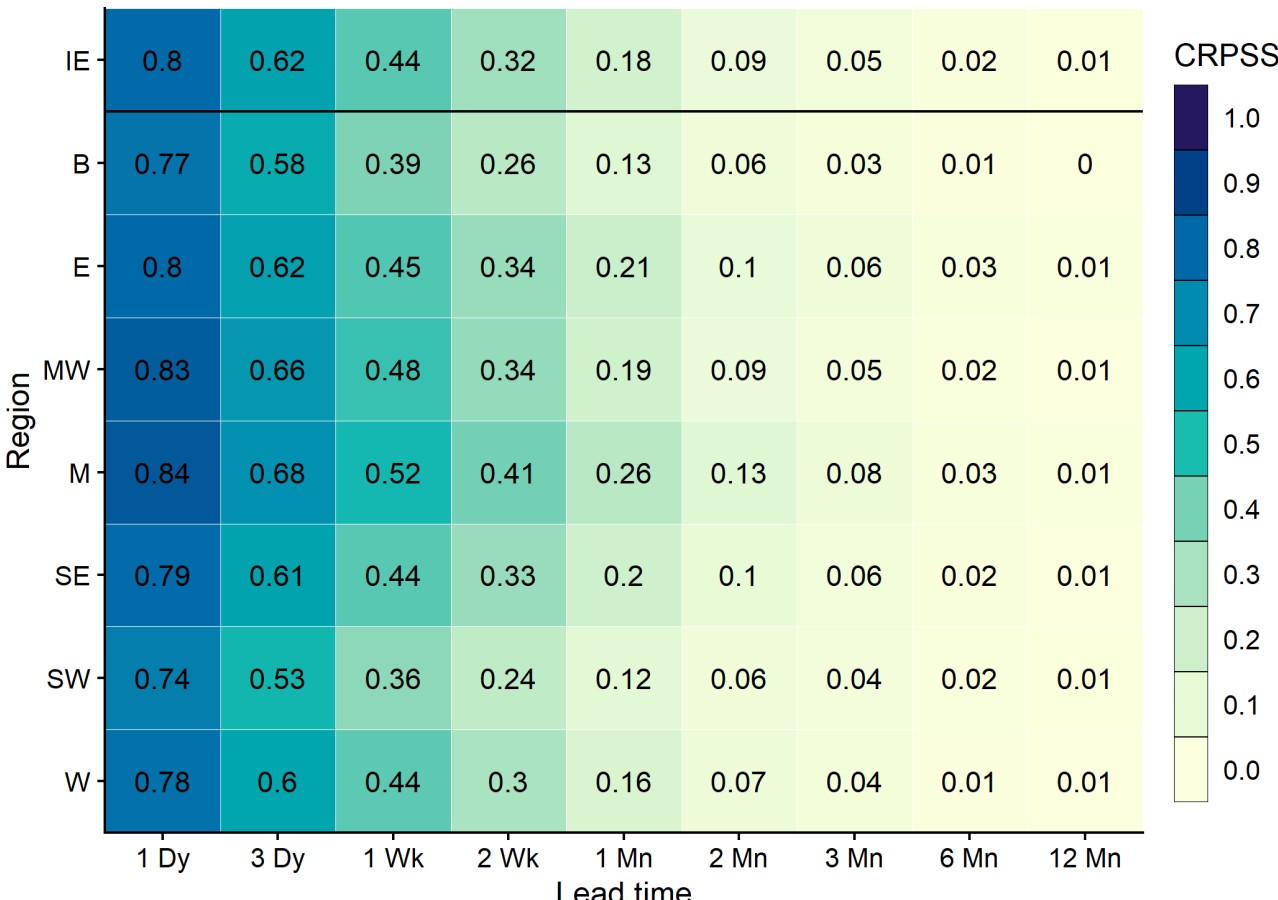

**Figure 5.** CRPSS values for Ireland (IE) and seven NUTS III regions (B, E, MW, M, SE, SW, and W) averaged across all initialisation months for a selection of lead times: short (1- and 3-day), extended (1- and 2-week), monthly (1- and 2-month), seasonal (3- and 6-month) and annual (12-month).

negative correlations with BFI and positive correlations with RBI and RR, highlighting that wetter catchments are more likely
to be those in which ESP has already been shown to perform poorly. Poor skill in these catchments is likely a combination of
high precipitation and low permeability which leads to more variable hydrological conditions as rainfall events propagate to
streamflow quickly. Finally, there are moderate negative correlations between ESP skill and S1085 ($\rho = -0.67$) and TAYSLO
($\rho = -0.59$) indicating that forecasts are less skilful in catchments with steeper gradients. Although these results are based on
the 1-month CRPSS averaged across all initialisation months, similar results are observed for a variety of different months and
lead times.



**Figure 6.** ESP skill for individual forecasts made at the 46 catchments for four sample lead times (columns) and four initialisation months (rows).

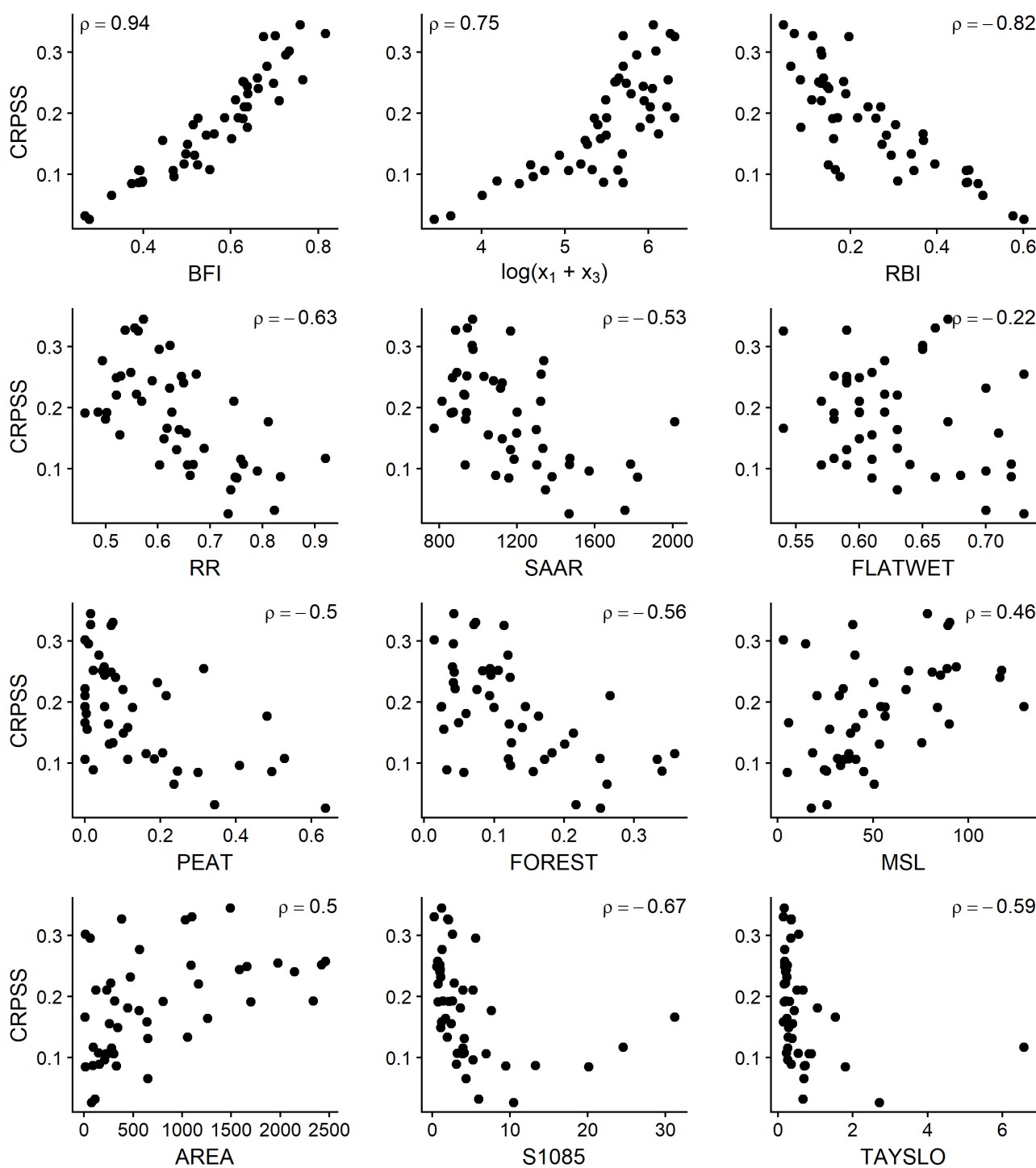

**Figure 7.** Relationship between 1-month ESP skill (CRPSS) and selected catchment descriptors.





## 3.5 Potential utility of ESP

The ability of ESP to discriminate between events and non-events is shown in Fig. 8 for low flow (lower tercile) and high flow (upper tercile) anomalies. In general, ESP is skilful at forecasting both low and high flow events up to 1 month ahead in the majority of catchments and for all initialisation months. Skilful forecasts of both event types are also possible at lead times

of 2 and 3 months, though to a lesser extent. These results highlight that ESP still has utility at longer lead times, even when overall performance as measured by the CRPSS may be low. However, this utility seldom extends beyond 3 months, except for specific catchments and initialisation dates, with little or no overall skill at lead times of 6 and 12 months. Some seasonality in skill is apparent. This is most notable at monthly lead times, where forecasts are most skilful in summer and least skilful in winter and spring, particularly December. ESP tends to be more skilful at predicting low flows than high flows.

## 3.6 Improvements in winter skill

The skill of NAO-conditioned ESP inputs is compared to historical ESP inputs using the CRPSS in Fig. 9. Whilst historical ESP is skilful in the majority of catchments at a 1-month lead time, there is a dramatic reduction in both the magnitude of skill and the number of catchments for which skilful forecasts can be made at 2- and 3-month lead times. NAO-conditioned ESP outperforms historical ESP relative to the climatology benchmark in all but one catchment at a 1-month lead time, though

these improvements are generally modest, with a median ($5^{th}$ and $95^{th}$ percentile) difference in CRPSS of 0.04 (0.007, 0.07). At a lead time of two months, NAO-conditioned ESP remains skilful against climatology in 98% of catchments, compared to historical ESP which is only skilful in 37% of catchments. The value of the NAO-conditioned ESP is more evident at a 3-month lead time, where skilful forecasts are still possible for several catchments in the Border and western regions, when historical ESP exhibits little or no skill across the majority of the sample.

Over the three lead times examined here, the greatest improvements are found for wet, fast-responding (low storage) catchments where the influence of initial conditions is weak, and predictability is more likely to be determined by prevailing meteorological conditions. For example, two of the best performing catchments for NAO-conditioned ESP are the Owenea (ID 38001) and the Fern (ID 39009). The Owenea has a BFI of 0.27, the lowest in the sample, with high SAAR (1753 mm), RR (0.82), and RBI (0.58) values. The Fern has a below-average BFI of 0.47 with similarly high SAAR and RR values of 1570 mm and 0.79,

respectively, although it is not as flashy (RBI = 0.18). It is notable that improvements in skill generally increase with lead time. This is likely due to the fact that the underlying NAO signal is not as strong over shorter averaging periods due to the noise of the individual weather systems. Moreover, only the seasonal mean NAO is re-scaled to account for the signal-to-noise problem when adjusting hindcasts, so skill is only present at the longer 3-month lead time. For example, at a 3-month lead time NAO-conditioned ESP improves forecast skill by ∼18% over historical ESP in both the Owenea and Fern, whereas gains of

7% and 12% are observed for 1- and 2-month lead times, respectively. These gains are comparable to those of (Beckers et al., 2016) using ENSO-conditioned ESP. NAO-conditioned forecasts generally perform the worst in slowly responding catchments with high baseflow contribution, as rainfall events do not immediately propagate into streamflow. In these catchments, initial







**Figure 8.** Distribution of ROC score values across all 46 study catchments for each initialisation month (January to December) and the same selection of lead times as in Fig. 5. Forecasts are considered skilful if the ROC score > 0.6.

conditions persist for longer and are the dominant source of ESP skill, thus there is little added benefit from sub-sampling precipitation and potential evaporation.

In addition to the CRPSS, the ROC score was also calculated for NAO-conditioned ESP. ROC scores for individual catchments and the full range of lead times are presented in Fig. 10. On average, NAO-conditioned ESP extends the lead time over which skilful forecasts can be made by 67% for low flows (1 month to 3 months) and 56% for high flows (40 days to 3 months). These are considerable improvements over historical ESP, which failed to meet the skill threshold in most catchments at longer





**Figure 9.** CRPSS values for historical ESP (left), NAO-conditioned ESP (middle), and the improvement made by NAO-conditioned ESP over historical ESP (right), at lead times of 1, 2, and 3 months.



lead times. For example, skilful predictions of low flows are possible in 78% of catchments at a 3-month lead time when using
NAO-conditioned ESP compared to only 13% of catchments when using historical ESP. This makes NAO-conditioned ESP
particularly effective at forecasting dry winters, which can be critical for water resources management. It is worth noting that
in many catchments NAO-conditioned ESP can 'lose' skill before later regaining it, with the ROC score falling only marginally
below the skill threshold. Although this is also observed for historical ESP, it is less frequent.

## 4  Discussion

### 4.1  When is ESP skilful

For short lead times (1–3 days), ESP forecasts are on average highly skilful (CRPSS $\geq$ 0.5) and for extended lead times (1–2
weeks) moderately skilful (CRPSS $\geq$ 0.25). Mean ESP skill decays rapidly with lead time. Hence, forecast skill for monthly,
seasonal, and annual lead times is on average much lower. This is because ESP relies on the long-term 'memory' of the
hydrological system. The cumulative effect of distinct meteorological forcing causes a divergence from the initial state that
grows with time. Thus, ESP suffers at longer lead times as there is little or no persistence of initial hydrological conditions.
Over longer periods, we find that ESP is most skilful out to a month ahead (CRPSS = 0.18) but that some predictability
(CRPSS > 0.05) is possible up to 3 months in advance. This rapid decline in forecast skill is consistent with findings from
several other benchmarking experiments, including Harrigan et al. (2018) and Girons Lopez et al. (2020), who noted a similar
deterioration in ESP skill in the UK and Sweden, respectively. Pechlivanidis et al. (2020) also reported a decline in seasonal
streamflow forecasting skill with increasing lead time across Europe. Persistence forecasts, which also rely on hydrological
memory as their main source of skill, have shown comparable results. For example, both Svensson (2016) and Foran Quinn
et al. (accepted) noted a reduction in the number of usable persistence forecasts in the UK and Ireland, respectively, when
moving from a 1-month forecast horizon to a 3-month forecast horizon.

ESP skill is also highly dependent on initialisation month. On average, at short-to-extended lead times (1 day to 2 weeks), ESP
is most skilful when initialised in summer and least skilful when initialised in winter. This is again consistent with previous
research, with higher predictability during dry seasons for forecasting methods that rely on hydrological memory reported for
the UK (Harrigan et al., 2018), Switzerland (Staudinger and Seibert, 2014), China (Yang et al., 2014), and parts of the Amazon
Basin (Paiva et al., 2012). This likely stems from a reduction in the direct contribution of precipitation to streamflow (Li
et al., 2009; Mo and Lettenmaier, 2014; Wood and Lettenmaier, 2008), which reduces variability and allows initial conditions
to persist for longer. In winter, lower evaporation rates lead to more effective rainfall which 'disrupts' the initial state and
limits the skill of ESP forecasts. This is particularly noticeable in flashy catchments with low baseflow contribution, where the
hydrological response is driven predominately by rainfall. Under such conditions, rainfall events propagate to streamflow at a
much faster rate and memory of initial conditions is lost quickly. At longer lead times, ESP is least skilful when initialised in
spring. Both Harrigan et al. (2018) and Svensson (2016) also found lower longer-range skill for forecasts initialised in spring in
the UK. The former attributed this to the transition from wet conditions with small soil moisture deficits to dry conditions with





**Figure 10.** Comparison of ROC scores achieved by historical ESP (left) and NAO-conditioned ESP (right) across all 46 study catchments and all lead times for low flow (lower tercile, a–b) and high flow (upper tercile, c–d) events.





large soil moisture deficits. Given that Ireland shares a similar precipitation regime to the UK, and that ESP skill is negatively impacted by high rainfall variability across the forecast period (Harrigan et al., 2018), this is also a plausible explanation for the results observed here.

### 4.2 Where is ESP skilful?

ESP is most skilful in the Midlands and least skilful in the Border and South-West. The Midlands is a lowland karst region which is underlain by permeable Carboniferous limestone, characterised by several locally and regionally important aquifers. Given that soils in this region are also well-drained, catchments located here have higher storage capacity and hence greater skill due to their long 'memory'. Both the Border and the West are poorly-drained regions, with the former characterised by unproductive bedrock aquifers. This partly explains the low storage capacity of catchments in these regions, which have quick
hydrological response times and poor persistence of initial conditions, resulting in lower ESP skill. Similar patterns were noted for persistence forecasts (Foran Quinn et al., accepted).

### 4.3 Why is ESP skilful?

ESP skill displays a strong relationship with modelled catchment storage capacity and catchment BFI values, with higher skill scores returned for catchments with greater storage. We conclude that storage capacity is primarily responsible for modulating
ESP skill. High BFI catchments have flow regimes dominated by slowly released groundwater (Chiverton et al., 2015) and are characterised by longer response times and lower streamflow variability (Sear et al., 1999; Broderick et al., 2016). This is conducive to greater persistence of initial conditions, with water storage in the soil creating a memory effect whereby anomalous conditions can take weeks or months to wane (Ghannam et al., 2014; Harrigan et al., 2018; Li et al., 2009). The role played by storage capacity is perhaps best illustrated by the fact that ESP skill decays at a much slower rate in catchments
with high BFI, especially during summer when streamflow is derived primarily from stored sources. For example, ESP is moderately skilful (CRPSS $\geq 0.25$) out a 2-month lead time for the Inny (ID 26021; BFI = 0.82) when initialised in July, but shows adequate (non-neutral) performance relative to climatology (CRPSS $> 0.05$) up to 4 months ahead. Moreover, whilst ESP tends to perform worse outside of summer months, catchments with relatively high SAAR but also high BFI yield above average skill scores in winter, spring, and autumn. In the Slaney (ID 12001; BFI = 0.71; SAAR = 1167 mm), skilful forecasts
are possible up to almost a year ahead in January and February, and up to 3–6 months ahead in spring and autumn. This likely stems from the delayed release of precipitation from groundwater stores (van Dijk et al., 2013), which can lead to temporal streamflow dependence for up to a season ahead (Chiverton et al., 2015).

### 4.4 Potential for operationalising ESP in Ireland

Our benchmarking results establish that ESP, in its traditional formulation, is skilful in a number of different scenarios, some-
times up to several months in advance. We recommend that ESP be used operationally in Ireland, similar to the HOUK (Prud-homme et al., 2017). Skilful streamflow forecasts at short to extended lead times could prove beneficial for water resources



management, particularly in areas such as Dublin where water supply systems have been operating close to capacity and face challenges of supply during dry periods. Given that the predictability of summer rainfall is notoriously difficult over northern Europe (Weisheimer and Palmer, 2014), the true utility of ESP may lie in its ability to leverage initial hydrological conditions,

particularly in high storage catchments, to skilfully predict streamflow up to a season ahead during dry months. Operationally, skill could be extended further by initialising forecasts more than once a month (e.g., Girons Lopez et al., 2020). As ESP has also been shown to accurately forecast both low and high flow events in many catchments up to at least a month in advance, it may also have practical relevance for decision-makers where it can act as an aid in the management of hydrologic extremes.

In the absence of skilful atmospheric forecasts or improved hydrological process representation, historical ESP provides a

lower limit of streamflow forecasting skill (Harrigan et al., 2018). However, we show that it is possible to improve ESP skill during winter by conditioning the method on the NAO. Improvements in forecast skill of 7–18% over lead times of 1 to 3 months are possible in catchments where meteorological conditions are the dominant control on skill. We do acknowledge, however, that these improvements are thus limited to specific catchments and are on top of a low initial skill base. In addition to improvements in overall forecast performance, NAO-conditioned ESP increases the lead time over which skilful forecasts of

low and high flows can be made. As winter is the most important season for groundwater recharge, during which reservoirs fill up to be used over the summer, the ability to accurately forecast wet and dry winters is extremely valuable for water managers, allowing them to anticipate the water situation beyond what is provided by the forecast alone. Hence, the greatest benefit of NAO-conditioned ESP may come from improved forecast discrimination rather than overall forecast performance, and it merits consideration for use in an operational setting.

## 4.5 Potential for future work

ESP skill is to a large extent dependent on the ability of hydrological models to accurately simulate catchment processes (Wang et al., 2011). It follows that further advances in ESP will likely require better representation of initial hydrological conditions and their evolution over time. Model structural and parameter uncertainty are therefore important considerations. Multi-parameter ensembles, data assimilation (e.g., Franz et al., 2014), state updating (e.g., Gibbs et al., 2018), and the use

of satellite data and remote sensing are potential ways through which estimates of initial conditions could be improved. It may also be possible to improve predictability by choosing model structures that are more capable of representing key flow pathways (i.e., groundwater, quick flow, etc.) and hence generate more accurate initial states. In this paper, GR4J is used as a parsimonious conceptual model to determine when and where skill is possible. Ongoing work will explore whether additional model complexity adds forecast skill at different initialisation and lead times through the use of models with different structures

and parameter dimensionality. In an operational setting, this could be extended to include more spatially discrete physically-based hydrological models that may better account for initial conditions. The additional benefit derived from using ensembles of models for maximising skill persistence could also be assessed for different lead times and initialisation months. This is a promising avenue, as model diversity has been shown to enhance forecast skill in ensemble experiments (Sharma et al., 2019).





We conducted a basic analysis of the relationship between forecast skill and catchment characteristics, using a small selection

of descriptors. A more comprehensive investigation of this relationship could be carried out, employing clustering techniques (e.g., Girons Lopez et al., 2020)Pechlivanidis-2020 and a wider range of hydrological signatures. As PCDs are available for a larger sample of 215 catchments, skill could be inferred in areas where modelling is not feasible (e.g., due to sparse or poor-quality observational data) based on *a priori* knowledge of local hydrological conditions. This could also be achieved by regionalising model parameters.

Finally, our use of NAO-conditioned ESP is only one way in which seasonal climate information can be incorporated into ESP forecasts. For example, another approach could be to condition model parameter sets instead of model inputs. It may also be possible to improve skill outside of winter, as the winter NAO has shown lagged correlations with summer rainfall over Ireland (Murphy et al., 2013) and river flows in the UK (Wilby, 2001). Seasonal forecasts of precipitation and temperature could also be incorporated directly into the process, in so-called climate-model based SHF (Yuan et al., 2015).

**5    Conclusions**

Ensemble Streamflow Prediction is a popular approach to seasonal hydrological forecasting that remains used some 40 years after first development. Here, we benchmarked ESP skill for a diverse sample of Irish catchments and conclude that it is skilful against streamflow climatology, but that the level of skill is strongly dependent on lead time, initialisation month, and individual catchment location and storage properties. In summary, we find that:

– ESP skill decays rapidly as a function of lead time, but the rate of decay is much slower in catchments with high storage capacity where initial conditions alone can provide skill up to several months in advance.

– For short (1–3 days), extended (1–2 weeks), and monthly lead times, ESP is most skilful when initialised during summer and least skilful when initialised during winter. At seasonal and annual lead times, ESP is least skilful when initialised during spring and about as skilful in autumn as it is in summer.

– ESP is most skilful in the Midlands, Mid-West, and East regions of Ireland, where slower responding catchments and the underlying lithology favour high storage capacity and longer hydrological 'memory.'

– ESP is capable of accurately forecasting both low and high flow events up to a month ahead in the majority of catchments. At lead times longer than 1 month, the number of catchments for which skilful forecasts can be made depends on initialisation month.

– NAO-conditioned ESP improves winter skill in fast-responding, low storage catchments in the Border and West regions, where the influence of meteorological forcing outweighs that from initial conditions. These improvements are more substantial over longer lead times of 2 and 3 months when the underlying NAO signal is less obscured by noise.





– NAO-conditioned ESP extends the lead times over which skilful forecasts of low and high flows can be made. This is particularly beneficial for forecasting dry winters, which can provide forewarning to water managers about potentially

problematic conditions.

We have demonstrated the skill of historical ESP for Ireland and highlighted its utility during the dry season, when demand for outlooks may be greatest. We have also shown how to improve ESP during winter, the season most critical for water managers. In light of the potential benefits for decision-makers, we recommend that ESP and conditioned ESP are operationalised, as they are serious contenders for producing skilful seasonal streamflow forecasts in Ireland.

*Data availability.* Streamflow data are available from the Office of Public Works (https://waterlevel.ie/) and the Environmental Protection Agency (https://www.epa.ie/hydronet/). Climate data and the ESP hindcast archive are available upon request from the authors. Supplement Table S1 includes metadata for all 46 catchments as well as model parameter values and data used to generate Table 2, Fig. 2, and Fig. 7.

### Appendix A: Non-parametric Kling–Gupta efficiency

The non-parametric Kling–Gupta efficiency ($KGE_{NP}$; Pool et al., 2018) is a modification of the traditional KGE (Gupta et al.,
2009) that uses the non-parametric Spearman rank correlation coefficient and normalised flow-duration curves to represent discharge dynamics and discharge variability, respectively. It is defined as:

$$KGE_{NP} = 1 - \sqrt{(\beta - 1)^2 + (\alpha_{NP} - 1)^2 + (\rho - 1)^2} \tag{A1}$$

$$\beta = \frac{\mu_s}{\mu_o} \tag{A2}$$

$$\alpha_{NP} = 1 - \frac{1}{2} \sum_{k=1}^{n} \left| \frac{Q_s(I(k))}{n \times \mu_s} - \frac{Q_o(J(k))}{n \times \mu_o} \right| \tag{A3}$$

Where $\rho$ is the non-parametric Spearman rank correlation coefficient between the simulated and observed time series, $\mu_s$ and $\mu_o$ are the mean of the simulated and observed time series, respectively, and $I(k)$ and $J(k)$ are the time steps when the $k^{th}$ largest
flow occurs within the simulated and observed time series, respectively. $\beta$ represents discharge volume. $\alpha_{NP}$ is calculated from the absolute difference between the normalised flow-duration curves.

### Appendix B: Continuous ranked probability skill score

The continuous ranked probability score (CRPS; Hersbach, 2000) measures the integrated squared difference between the forecast cumualtive distribution function (CDF) and the empirical CDF of the observation. For a continuous random variable
$X$ (e.g., streamflow) with probability density function $f_X$, the CRPS between the forecast CDF, denoted $F_X$, and the empirical CDF of the observation $y$, denoted $F_y$, is defined as:





$$\mathrm{CRPS}(F_X, y) = \int_{-\infty}^{\infty} [F_X(x) - F_y(x)]^2 \, dx \tag{B1}$$

$$F_X(x) = \int_{-\infty}^{x} f_X(t) \, dt \tag{B2}$$

$$F_y(x) = H(x - y) \tag{B3}$$

Where H is the Heaviside step function: $H(x) = 1$ for $x \geq 0$ and $H(x) = 0$ for $x < 0$. The continuous ranked probability skill score (CRPSS) is then given by:

$$\mathrm{CRPSS} = 1 - \frac{\overline{\mathrm{CRPS_{Sys}}}}{\overline{\mathrm{CRPS_{Ref}}}} \tag{B4}$$

Where $\overline{\mathrm{CRPS_{Sys}}}$ is the average CRPS of the forecasting system for a set of forecast–observation pairs, and $\overline{\mathrm{CRPS_{Ref}}}$ is the equivalent for the reference forecast. The CRPSS ranges from low negative values to 1, with positive (negative) values indicating better (worse) performance than the reference forecast.

*Author contributions.* SD designed the study with input from SH. JK, AAS, and NS contributed the GloSea5 data used to condition the
ESP method. SD carried out the modelling, analysed the results, and produced the figures under the supervision of CM. SD prepared the
manuscript with contributions from all co-authors.

*Competing interests.* The authors declare no competing interests.

*Acknowledgements.* CM, SD, SG, and DFQ gratefully acknowledge funding from Science Foundation Ireland (Grant No. SFI/17/CDA/4783).





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
