# Peer review of "Conditioning Ensemble Streamflow Prediction with the North Atlantic Oscillation improves skill at longer lead times"

_Hydrology and Earth System Sciences, 2020_

## Referee Comment (RC1) · Anonymous Referee #1 · 11 Jan 2021

The authors apply the ensemble streamflow prediction (ESP) and conditioned ESP to quantify the predictability across time scales over different catchments in Ireland by hydrological model. In this work, they find that the prediction based on memory of initial hydrological condition is skillful up to several months, especially in summer. In addition, the skillful prediction of North Atlantic Oscillation (NAO) is benefit to the hydrological prediction in winter. Overall, this manuscript is well prepared and organized. I only have the following minor suggestions.

1. Besides the CRPSS, NSE and ROC, the correlation coefficient (CC) is an important index. Therefore, I think you should also provide the CC in your analysis, such as the

relationship between observation and simulation.

2. Why do you divide the 46 catchments into 8 regions? The Figure 5 also can be plotted as the Figure 10, or the Figure 10 can be arranged as the Figure 5?

3. You should explain the conditioned ESP in more details to ensure reproducibility. For example, you didn't explain that what the '17' is in the Line 193. In your work, the conditioned ESP is able to improve the skill significantly over many catchments in Ireland. In addition to conditioned ESP, the post-ESP is another prediction method, which involves the information from initial hydrological condition and internal climate variability as well (Yuan & Zhu, 2018). You can compare the impacts of these two methods on the improving of prediction skill.

4. Your work represents the skill of conditioned ESP performs better than the ESP over many catchments. However, the information of NAO reduces the skill in a few regions where the ESP is skillful, especially at 3-month lead, such as the three catchments in East region. This is an interesting phenomenon, you should discuss it.

5. Many previous works show that the memory of initial hydrological condition in winter is more important because the snow cover plays a key role in seasonal streamflow forecast. However, you get a different conclusion in the work. You should discuss the difference.

6. Each panels in your figures should be labeled.

Yuan, X., & Zhu, E. (2018). A first look at decadal hydrological predictability by land surface ensemble simulations. Geophysical Research Letters, 45(5), 2362–2369. https://doi.org/10.1002/2018GL077211

―――――――――――――――――――――

---

## Referee Comment (RC2) · Anonymous Referee #2 · 14 Jan 2021

This paper has presented an investigation of ensemble streamflow prediction (ESP) for 46 catchments in Ireland. The GR4J model is employed to formulate the rainfall-runoff relationship and perform streamflow forecasting. The forecast skill is evaluated and then related to a range of catchment attributes, e.g., base flow index, flashiness index, and runoff ratio. The results show that skillful forecasts are generated using ESP and that the skill can be attributed to catchment attributes and North Atlantic Oscillation (NAO). Overall, the paper is well-written with the methods and results clearly presented.

There are a few comments for further improvements of the paper.

**HESSD**

First, reliability is an important feature of ensemble forecasts. Specifically, reliability indicates the agreement between forecast probability and mean observed frequency. For streamflow forecasting, attention is usually paid to high- and low-flow events. Therefore, it would be meaningful to show whether ensemble forecasts generated by ESP yield reliable probabilistic forecasts of high- and low-flow events at different lead times. For more information on forecast reliability, please refer to https://www.cawcr.gov.au/projects/verification/#:~:text=If%20we%20take%20the%20term,the%20quality%20of%20a%20f

Second, there recently is an interesting paper on the influence of NAO on flooding and drought over Europe (Changes in North Atlantic atmospheric circulation in a warmer climate favor winter flooding and summer drought over Europe, E Rousi, F Selten, S Rahmstorf, D Coumou, Journal of Climate, 2020). This paper can offer some climatological insights when relating forecast skill to NAO.

---

## Author Response (AR1)

**Response to the Editor**

Dear Authors,

I would like to thank your responses. It would be useful to upload the revised manuscript, and a version with tracked changes. BTW, although sticking to the probabilistic evaluation is ok, the comments on CC might be biased. It is a commonly used metric for assessing predictive skill both in meteorology and hydrology.

Looking forward to your revision.

Regards
Xing Yuan

Dear Xing,

Thank you for inviting us to revise our manuscript. We sincerely apologise for the delay and hope that our revisions are satisfactory. We are sticking to the probabilistic evaluation, though we disagree with the contention that our comments on the use of the correlation coefficient are biased. Please find below our original response to referee comments with relevant changes in the manuscript indicated in red text. Line numbers refer to the tracked changes document.

Regards,
Seán Donegan

**Response to Anonymous Referee #1**

Referee comments are labelled consecutively (e.g., R#1-1 is comment 1) and given in blue text.

The authors apply the ensemble streamflow prediction (ESP) and conditioned ESP to quantify the predictability across time scales over different catchments in Ireland by hydrological model. In this work, they find that the prediction based on memory of initial hydrological condition is skillful up to several months, especially in summer. In addition, the skillful prediction of North Atlantic Oscillation (NAO) is benefit to the hydrological prediction in winter. Overall, this manuscript is well prepared and organized. I only have the following minor suggestions.

We thank the referee for their constructive review of our manuscript. Please find our point-by-point response below.

**R#1-1.** Besides the CRPSS, NSE and ROC, the correlation coefficient (CC) is an important index. Therefore, I think you should also provide the CC in your analysis, such as the relationship between observation and simulation.

We recognise that the correlation coefficient (CC) is a common verification metric in hydrometeorological forecasting; however, we do not believe a deterministic analysis of this nature is necessary in a paper whose overarching focus is the evaluation of a probabilistic forecasting system. Our choice of the continuous ranked probability skill score (CRPSS) was informed by the ensemble forecasting literature. It is for most cases the recommended evaluation method for ensemble hydrological forecasts and it features prominently in official reports and peer-reviewed research (Pappenberger et al., 2015). Furthermore, it is well-known that fundamental deficiencies in the CC limit its usefulness for evaluating forecasts. Among these are its sensitivity to outliers and its insensitivity to additive and proportional differences between model predictions and observed data (Moriasi et al., 2007). This may lead to a substantial overestimation of forecast performance (Murphy, 1988). To illustrate this, we conducted an assessment of ESP skill using the CC. However, as the CC and the CRPSS are not directly comparable, we also calculated the mean absolute error skill score (MAESS) as an additional measure of deterministic performance. The MAESS is the equivalent of the CRPSS for a single-valued forecast and differences between these metrics can therefore be attributed to the use of the probabilistic ensemble. Our findings are presented in Fig. S1 (below).

[Figure]

**Figure S1.** Comparison of mean CRPSS, MAESS, and CC values at each lead time across: (a) all catchments and initialisation months, and (b) all catchments for each of 12 initialisation months, January to December.

Whilst the CRPSS and the MAESS are almost indistinguishable, the CC is systematically higher (lower) than both for skilful (unskilful) forecasts. For example, at a 1-month lead time, across all catchments and initialisation months, mean (5[th] and 95[th] percentile, not shown) CRPSS is 0.18 (0.03, 0.42), MAESS is 0.18 (0.02, 0.39), and CC is 0.5 (0.18, 0.75). Similar is observed for individual initialisation months. A comparison of the CRPSS and MAESS shows, however, that it cannot be concluded based simply on a high (low) CC alone that the ensemble mean is more (less) skilful than the full ensemble. Indeed, it is possible to have a forecast that is highly correlated with the observations, but with sufficiently severe bias that it has no practical use (Wilks, 2019). It is also worth noting that the choice of metric does not change our conclusions regarding the spatiotemporal distribution of ESP skill.

In light of the above, we would prefer to omit the CC from our analysis for the following reasons. First, it does not reveal anything about forecast performance, temporally or spatially, that is not already evident from the other verification metrics employed. Second, we would like to avoid potential misevaluation of forecast skill arising from differences in magnitude between the CRPSS and CC. Finally, an unbiased CC analysis would require us to detrend all of our time series. We are unsure what effect, if any, this would have on the results from persistence-based forecasting methods such as ESP.

As indicated, we prefer not to include the correlation coefficient in our paper. We have added analysis on forecast reliability using the PIT score as requested by R#2. This has extended our manuscript and we feel that adding the CC would increase the complexity of our work which we believe has been significantly improved by the review process.

**R#1-2.** Why do you divide the 46 catchments into 8 regions? The Figure 5 also can be plotted as the Figure 10, or the Figure 10 can be arranged as the Figure 5?

We divide the 46 catchments into 7 regions ('IE' in Fig. 5 is the whole of Ireland and included only for comparison purposes) to facilitate spatial analysis and aid in the interpretation of the results. A key research aim (p. 4, L#88) was to identify where ESP was skilful, at both regional and catchment scales; however, we also wanted to investigate if performance differed between regions with contrasting hydroclimate characteristics. This latter point is not explicitly stated and will be made clear in the final version of the manuscript. We used the European Union's NUTS III regions both for consistency with Foran Quinn et al. (2021), who employed the same designation in their evaluation of persistence forecasts for Ireland, but also because no work has been done to create clearly defined Irish hydroclimate regions. Whilst the NUTS III regions do not inherently lend themselves to hydrological analysis, we

note that grouping the catchments in this way did yield regions that were diverse in terms of their hydrological and climatological characteristics. This is clear from Table 2, but we propose amending Sect. 2.1 to make this more apparent by including summary statistics directly in the text.

We would prefer to keep Fig. 5 and Fig. 10 as they currently are. The purpose of Fig. 5 is to provide a broad overview of regional skill. Hence, we plot only regional averages for a selection of key lead times commonly used in seasonal forecasting. Redrawing Fig. 5 with all catchments and/or lead times would create an unnecessarily complex figure not suitable for what we are trying to convey in this section. Furthermore, the spatial distribution of skill at the individual catchment scale is already covered in Fig. 6 for specific lead times and initialisation months. Figure 10 supports a key conclusion of the paper: that NAO-conditioned ESP extends the lead time over which skilful forecasts of low and high flows can be made and is thus a useful tool for predicting anomalously wet or dry winters. We feel rearranging Fig. 10 to be similar to Fig. 5 would downplay how significant these improvements are.

We have revised the manuscript to clarify our decision to divide the catchments into different regions (L#108–14).

**R#1-3.** You should explain the conditioned ESP in more details to ensure reproducibility. For example, you didn't explain that what the '17' is in the Line 193. In your work, the conditioned ESP is able to improve the skill significantly over many catchments in Ireland. In addition to conditioned ESP, the post-ESP is another prediction method, which involves the information from initial hydrological condition and internal climate variability as well (Yuan & Zhu, 2018). You can compare the impacts of these two methods on the improving of prediction skill.

To condition the ESP method, we begin with a 51-member ensemble of raw NAO predictions from GloSea5. These predictions consist of monthly NAO values for each winter (DJF) period between 1993/94 and 2015/16. To remove the signal-to-noise discrepancy found in the raw ensemble, the predictions are adjusted following the method of Stringer et al. (2020). The adjusted monthly values are used to select 10 non-sequential DJF analogues (e.g., December 2007, January 1980, February 2011) where the mean observed seasonal NAO approximates the mean adjusted seasonal NAO hindcast. This yields a 510-member ensemble of analogue date sequences which are then used to sample observed precipitation and potential evaporation for input to the hydrological model. We believe this is adequately explained in the text; however, we will make it clear that readers should refer to Stringer et al. (2020) for a more detailed description of the adjustment procedure and the selection of analogue dates.

The 51-member GloSea5 ensemble is a lagged ensemble created by combining three separate initialisations each of which has 17 ensemble members. This is what the '17' on L#193 refers to. For clarity, we will change L#192–3 to the following: "For each DJF period between 1993–2015, we combined GloSea5 hindcasts initialised on 1, 9, and 17 November, each with 17 ensemble members, to create a 51-member lagged ensemble of raw NAO predictions."

We thank the referee for suggesting a comparison of NAO-conditioned ESP, as we present it in this manuscript, and a post-processed ESP similar to that used by Yuan and Zhu (2018). This is an important area of research and such comparisons contribute much needed understanding as to the most effective methods for incorporating seasonal climate information into hydrological forecasts. However, we believe it is ultimately outside the scope of this paper and should instead be addressed in future work. Nevertheless, we would like to acknowledge Yuan and Zhu (2018) in our manuscript. Whilst differences in study design do not allow for a direct comparison of results, we will include a reference to this paper in our introduction (p. 3, para. 3) as an additional example of how climate information can be used to improve ESP forecasts. We will also edit Sect. 4.5 to highlight this avenue for further research suggested by the referee.

We have revised the manuscript to clarify that we are using a lagged ensemble (L#206–8) and we now direct interested readers to Stringer et al. (2020) for more details on the hindcast adjustment procedure and the selection of analogue dates (L#217–18). We have added a reference to Yuan and Zhu (2018) in our introduction (L#77–8) and updated Sect. 4.4 to note post-processed ESP as an alternative to the conditioning approach we used here (L#509–13).

**R#1-4.** Your work represents the skill of conditioned ESP performs better than the ESP over many catchments. However, the information of NAO reduces the skill in a few regions where the ESP is skillful, especially at 3-month lead, such as the three catchments in East region. This is an interesting phenomenon, you should discuss it.

We agree that this is an interesting phenomenon. Reductions in skill from NAO-conditioned ESP are observed for 12 catchments in total, and only at a 3-month lead time. We feel it is important to note, however, that these reductions are very minor and that the performance of both historical and conditioned ESP at this lead time can be defined within the limits of what Bennett et al. (2017) refer to as 'neutral skill' (±0.05 CRPSS). Thus, despite some degradation in the CRPSS, the NAO-conditioned ESP does not perform considerably worse than the historical ESP overall. We observe that skill decreases with lead time in these catchments, whereas skill was shown to increase with lead time in others. We believe this is due to differences in the relative contribution of initial hydrological conditions and meteorological forcing to ESP skill. Of the 12 catchments considered here, the majority of them are

characterised by high baseflow contribution (BFI > 0.5) and long recession times. Hence, hydrological response is controlled predominantly by the slow release of water from reservoirs and initial conditions act as the primary source of skill. The combination of initial conditions and subsampled climate information grants modest improvements in skill up to a 1-month lead time. However, at longer lead times, improved atmospheric representation alone cannot compensate for divergences from the initial state. Skill deteriorates as a result, eventually becoming negative. We see the opposite in flashy catchments with low storage capacity, where rainfall events propagate to streamflow at a much faster rate and memory of initial conditions is lost quickly. Knowledge of meteorological forcing plays a more important role and the greatest benefits from conditioning the ESP method emerge at longer lead times when the NAO signal is less obscured by noise. It would be possible to confirm this by quantifying the contribution of each source of skill using an ESP and reverse ESP approach (e.g., Wood and Lettenmaier, 2008), but that is outside of the scope of the present study. We will include this additional discussion in Sect. 3.6 in the final version of the manuscript.

We have restructured Sect. 3.7 to address this degradation in performance and more clearly discuss the relative contribution of initial conditions and meteorological forcing to forecast skill (L#358–84).

**R#1-5.** Many previous works show that the memory of initial hydrological condition in winter is more important because the snow cover plays a key role in seasonal streamflow forecast. However, you get a different conclusion in the work. You should discuss the difference.

Snow is not known to make a substantial contribution to precipitation in Ireland. To illustrate this, we adopted a simple method from Berghuijs et al. (2014) to calculate the mean fraction of precipitation falling as snow ($\overline{F_s}$) for each catchment between 1992 and 2017. Precipitation on days with an average temperature below 1°C is considered to be entirely snowfall, whereas on days with an average temperature greater than or equal to 1°C, precipitation is considered to be entirely rainfall. No catchments were found to have a significant amount of snowfall ($\overline{F_s} > 0.15$), with a median (5[th] and 95[th] percentile) $\overline{F_s}$ of 0.008 (0.004, 0.01) across the sample. We will include this information in the final version of the manuscript and explicitly state that snow is not a major driver of hydrological response in Ireland. Snow cover and snow melt therefore do not play a significant role in seasonal streamflow forecasting in an Irish context.

We have added this information to Sect. 2.1 (L#134–8) and to Table 2 and Table S1. Please note that the values of $\overline{F_s}$ given in the original response were incorrect due to an error in the calculation. This has been fixed in the manuscript.

We will ensure all figures are appropriately labelled in the final version of the manuscript.

We have redrawn our figures to improve overall presentation and consistency. We have used faceting to make clear what is shown in each figure panel.

**Response to Anonymous Referee #2**

Referee comments are labelled consecutively (e.g., R#2-1 is comment 1) and given in blue text.

This paper has presented an investigation of ensemble streamflow prediction (ESP) for 46 catchments in Ireland. The GR4J model is employed to formulate the rainfall-runoff relationship and perform streamflow forecasting. The forecast skill is evaluated and then related to a range of catchment attributes, e.g., base flow index, flashiness index, and runoff ratio. The results show that skillful forecasts are generated using ESP and that the skill can be attributed to catchment attributes and North Atlantic Oscillation (NAO). Overall, the paper is well-written with the methods and results clearly presented. There are a few comments for further improvements of the paper.

We thank the referee for their constructive review of our manuscript. Please find our point-by-point response below.

**R#2-1.** First, reliability is an important feature of ensemble forecasts. Specifically, reliability indicates the agreement between forecast probability and mean observed frequency. For streamflow forecasting, attention is usually paid to high- and low-flow events. Therefore, it would be meaningful to show whether ensemble forecasts generated by ESP yield reliable probabilistic forecasts of high- and low-flow events at different lead times.

We agree that reliability is an important consideration when evaluating ensemble hydrological forecasts. To address this, we will include an additional verification metric, the probability integral transform (PIT) diagram (Gneiting et al., 2007) as a means of assessing the reliability of high- and low-flow forecasts. The PIT diagram is the cumulative distribution of the PIT values, which measure the position of the observations relative to the forecast distribution. For a perfectly reliable forecast, the observations will fall uniformly within the forecast distribution and the PIT diagram will correspond to the 1-to-1 diagonal. Forecasts that systematically under- (over-) predict will have a PIT diagram below (above) the diagonal, whereas under- (over-) dispersive forecasts will have a transposed S-shaped (S-shaped) PIT diagram (Arnal et al., 2018). For comparison on large datasets, the area between the PIT diagram and the 1-to-1 diagonal can be calculated to provide a numerical measure of reliability. This can further be converted to a skill score for ease of interpretation (Arnal et al., 2018; Crochemore et al., 2017). We believe incorporating the PIT diagram into our manuscript will complement our use of the ROC score and help improve our overall analysis.

We have added reliability analysis to our manuscript (Sect. 2.4.2, Sect. 3.5, Sect. 3.7) as requested by the referee. This has been a useful addition, thanks for the recommendation.

**R#2-2.** Second, there recently is an interesting paper on the influence of NAO on flooding and drought over Europe (Changes in North Atlantic atmospheric circulation in a warmer climate favor winter flooding and summer drought over Europe, E Rousi, F Selten, S Rahmstorf, D Coumou, Journal of Climate, 2020). This paper can offer some climatological insights when relating forecast skill to NAO.

Thank you for bringing this paper to our attention. We will review its contents and revise our manuscript accordingly to include any additional insight it may provide.

We would like to thank the referee again for recommending this paper. However, we did not see a direct relevance beyond the papers already included. It is certainly useful to other research we are doing.

**References**

Arnal, L., Cloke, H. L., Stephens, E., Wetterhall, F., Prudhomme, C., Neumann, J., Krzeminski, B., and Pappenberger, F.: Skilful seasonal forecasts of streamflow over Europe?, Hydrol. Earth Syst. Sci., 22, 2057–2072, https://doi.org/10.5194/hess-22-2057-2018, 2018.

Bennett, J. C., Wang, Q. J., Robertson, D. E., Schepen, A., Li, M., and Michael, K.: Assessment of an ensemble seasonal streamflow forecasting system for Australia, Hydrol. Earth Syst. Sci., 21, 6007–6030, https://doi.org/10.5194/hess-21-6007-2017, 2017.

Berghuijs, W. R., Woods, A., and Hrachowitz, M.: A precipitation shift from snow towards rain leads to a decrease in streamflow, Nat. Clim. Change, 4, 583–586, https://doi.org/10.1038/nclimate2246, 2014.

Crochemore, L., Ramos, M.-H., Pappenberger, F., and Perrin, C.: Seasonal streamflow forecasting by conditioning climatology with precipitation indices, Hydrol. Earth Syst. Sci., 21, 1573–1591, https://doi.org/10.5194/hess-21-1573-2017, 2017.

Foran Quinn, D., Murphy, C., Wilby, R. L., Matthews, T., Broderick, C., Golian, S., Donegan, S., and Harrigan, S.: Benchmarking seasonal forecasting skill using river flow persistence in Irish catchments, Hydrolog. Sci. J., https://doi.org/10.1080/02626667.2021.1874612, 2021.

Gneiting, T., Balabdaoui, F., and Raftery, A. E.: Probabilistic forecasts, calibration and sharpness, J. R. Statist. Soc. B, 69, 243–268, https://doi.org/10.1111/j.1467-9868.2007.00587.x, 2007.

Moriasi, D. N., Arnold, J. G., Van Liew, M. W., Bingner, R. L., Harmel, R. D., and Veith, T. L.: Model evaluation guidelines for systematic quantification of accuracy in watershed simulations, Trans. ASABE, 50, 885–900, https://doi.org/10.13031/2013.23153, 2007.

Murphy, A. H.: Skill Scores Based on the Mean Square Error and Their Relationships to the Correlation Coefficient, Mon. Weather Rev., 116, 2417–2424, https://doi.org/10.1175/1520-0493(1988)116<2417:SSBOTM>2.0.CO;2, 1988.

Pappenberger, F., Ramos, M.-H., Cloke, H. L., Wetterhall, F., Alfieri, L., Bogner, K., Mueller, A., and Salamon, P.: How do I know if my forecasts are better? Using benchmarks in hydrological ensemble prediction, J. Hydrol., 522, 697–713, https://doi.org/10.1016/j.jhydrol.2015.01.024, 2015.

Stringer, N., Knight, J., and Thornton, H.: Improving Meteorological Seasonal Forecasts for Hydrological Modeling in European Winter, J. Appl. Meteorol. Climatol., 59, 317–332, https://doi.org/10.1175/JAMC-D-19-0094.1, 2020.

Wilks, D. S.: Statistical Methods in the Atmospheric Sciences, 4th Edition, Elsevier, Amsterdam, 2019.

Wood, A. W. and Lettenmaier, D. P.: An ensemble approach for attribution of hydrologic prediction uncertainty, Geophys. Res. Lett., 35, L14401, https://doi.org/10.1029/2008GL034648, 2008.

Yuan, X. and Zhu, E.: A first look at decadal hydrological predictability by land surface ensemble simulations, Geophys. Res. Lett., 45, 2362–2369, https://doi.org/10.1002/2018GL077211, 2018.